# Obesity and Associated Factors in Brazilian Adults: Systematic Review and Meta-Analysis of Representative Studies

**DOI:** 10.3390/ijerph21081022

**Published:** 2024-08-02

**Authors:** Cecilia Alcantara Braga Garcia, Karina Cardoso Meira, Alessandra Hubner Souza, Ana Laura de Grossi Oliveira, Nathalia Sernizon Guimarães

**Affiliations:** 1Faculty of Medical Sciences of Minas Gerais, Belo Horizonte 30130-110, MG, Brazil; ceciliaabgarcia@gmail.com (C.A.B.G.); alessandra.souza@cienciasmedicasmg.edu.br (A.H.S.); 2School of Health, Universidade Federal do Rio Grande do Norte, Natal 59078-970, RN, Brazil; karina.meira@ufrn.br; 3Graduate Program in Infectious Diseases and Tropical Medicine, Universidade Federal de Minas Gerais, Belo Horizonte 30130-100, MG, Brazil; analauragrossi@gmail.com; 4Department of Nutrition, School of Nursing, Universidade Federal de Minas Gerais, Belo Horizonte 30130-100, MG, Brazil

**Keywords:** obesity, prevalence, adults, Brazil, systematic review, meta-analysis

## Abstract

To review the literature and select population-based studies that are representative of Brazilian capitals or Brazil as a whole to estimate the prevalence of obesity among Brazilian adults. The Preferred Reporting Items for Systematic Reviews and Meta-Analyses (PRISMA) were used. The search was conducted in six databases and reference lists of included studies. We included observational studies but excluded interventional studies, reviews, in vitro studies, and editorials. The study population consisted of young adults and adults (18 to 59 years old). Adolescents, infants, children, the elderly, and pregnant women were excluded. The primary outcomes were the prevalence of obesity among Brazilian adults, both men and women. The secondary outcomes were factors associated with obesity. The meta-analysis was performed using Rstudio software, version 4.1.0, by using the ‘Meta’ package, version 5.0-0. The search strategy identified 5634 references, of which 19 studies and 21 national surveys were included in the meta-analysis. The pooled prevalence of obesity in Brazilian adults was 20.0% (95% CI: 14.0–25.0%) while in the capitals it was 17.0% (95% CI: 16.0–19.0%). Across the regions of Brazil, the prevalence ranged from 11.0% to 17.0%, with the highest frequency in the south. Increases in obesity prevalence were observed for both sexes in almost all periods, with consistently higher rates among women in most cases. The prevalence of obesity among Brazilian adults is high, with no statistically significant differences found in the subgroup analysis.

## 1. Introduction

Obesity is a critical public health issue and a complex multifactorial disease that has reached pandemic proportions. Obesity rates have increased across all ages and both sexes, regardless of geographical location, ethnicity, or socioeconomic status [1,2,3].

According to estimates by the World Health Organization (WHO), by 2025, there will be more than 700 million obese adults. An analysis conducted by the non-communicable chronic diseases Risk Factor Collaboration (NCD-RisC) group, which included 19.2 million people, projected that by 2025, the global prevalence of obesity will reach 18% among men and over 21% among women, with severe obesity rates potentially exceeding 9% among women and 6% among men [4]. The global prevalence of obesity is predicted to rise from 14% in 2020 to 24% by 2035 [5].

In Latin America and the Caribbean, obesity is a growing problem, with the Americas region having the highest prevalence of obesity in the world [6]. According to the Pan American Health Organization (PAHO) Regional Panorama of Food and Nutrition Security 2021, the prevalence of obesity in adults over 18 years of age in 2016 was 24.2%, significantly above the global average. Between 2000 and 2016, there was a significant increase in obesity prevalence, with a rise of 9.5 percentage points in the Caribbean, 8.2 percentage points in Mesoamerica, and 7.2 percentage points in South America [7].

In Brazil, the Telephone-based Surveillance of Risk and Protective Factors for Chronic Diseases (VIGITEL), organized by the Ministry of Health, reported a 96% increase in obesity prevalence over the past 15 years, rising from 11.8% in 2006 to 22.4% in 2021 [8].

A high body mass index (BMI) is a significant risk factor for non-communicable chronic diseases (NCDs), mental health changes such as depression, and impaired quality of life. Several factors contribute to the obesity epidemic worldwide, including sociodemographic, economic, environmental, physiological, and psychosocial factors, particularly in Latin America and Brazil. Changes in dietary patterns, increased sedentary behavior, and urbanization are notable contributors to the rise in obesity rates.

NCDs associated with inadequate nutrition have a significant direct and indirect impact on mortality in Brazil, accounting for 71% of all deaths. The four main groups of NCDs linked to poor nutrition were responsible for 55% of all deaths in Brazil [9]. NCDs were the leading cause of premature death (ages 30 to 69) for both women and men, representing 37% of deaths in the 30–49 age group and 65% in the 50–69 age group [9].

Between 2011 and 2022, the Plan to Combat NCDs was carried out, which provided for 12 goals with the aim of promoting the development and implementation of effective, integrated, sustainable, and evidence-based public policies for the prevention and control of NCDs along with their risk factors, such as obesity, in addition to supporting health services aimed at chronic diseases. Currently, the Plan to Combat Chronic Diseases and Non-Communicable Diseases in Brazil is in force, which will cover the period from 2021 to 2030. To reach the goals, 226 strategic actions were defined to be developed by the Ministry of Health, by the states, by the Federal District, and by the municipalities. Actions include reducing alcoholism and smoking, proper and healthy eating, and physical activity [10].

Given the personal, social, and economic impact, synthesizing the evidence on the prevalence of obesity is crucial. This can aid health policy managers in better understanding the scope of this chronic disease and in developing strategies to improve access to treatment for individuals. It also highlights the need for investing in preventive measures, especially in emerging countries like Brazil, which faces significant territorial and regional inequalities [11].

In this context, we propose to systematically review the literature, selecting population-based studies representative of Brazilian capitals or Brazil as a whole to estimate the prevalence of obesity among Brazilian adults.

## 2. Materials and Methods

### 2.1. Protocol and Registration

This systematic review and meta-analysis followed the Cochrane Guidelines for Systematic Reviews of Interventions and was written according to the Preferred Reporting Items for Systematic Reviews and Meta-Analyses (PRISMA) [12,13]. The study protocol was registered on PROSPERO (#CRD42023390871).

### 2.2. Search Strategy

To answer the question, “What is the prevalence of obesity in Brazilian adults by macro-regions?”, we searched six independent databases to perform the sensitive literature search: MEDLINE, Embase, Web of Science, Central (by Cochrane Library), Scopus, and the Latin American and Caribbean Health Sciences Literature (LILACS). Additionally, we hand-searched the reference lists of the included studies. Based on the population surveys included, the SIDRA platform was consulted.

There were no restrictions based on language, date, document type, or publication status for including records. The last search was conducted in January 2023. Descriptors were identified in Medical Subject Headings (MeSH), Descritores em Ciências da Saúde (DeCS), and Embase Subject Headings (Emtree). Later, they were combined with the boolean operator AND, whereas their synonyms were combined with the boolean operator OR. The following meshes formed the herein-used search strategy, which was adapted based on descriptors in each database: “prevalence”, “obesity”, and “Brazil”. The search strategy adopted in each database is presented in the Appendix A.

### 2.3. Outcomes

The primary outcomes were obesity prevalence in Brazilian adults, obesity prevalence in Brazilian men, and obesity prevalence in Brazilian women.

The secondary outcomes were factors associated with obesity, such as sociodemographic factors (sex, age, race/skin color, level of education, urban or rural housing, and distribution in Brazil’s macro-regions), socioeconomic factors (health insurance holder), clinical factors (noncommunicable diseases and self-assessments of health condition), and modifiable risk factors (smoking, alcoholism, intake of ultra-processed foods, and physical activity). The description of the sample design of the primary studies can be found in the Appendix A.

### 2.4. Eligibility Criteria

We included observational studies (cross-sectional, case–control, and cohort studies). Interventional studies, reviews, in vitro studies, and editorials were excluded. Participants/population were young adults and adults (18 to 59 years old). Adolescents, infants, children, the elderly, and pregnant women were excluded.

### 2.5. Study Selection and Data Extraction

Electronic search results from defined databases were uploaded to the Rayyan Qatar Computing Research Institute [14].

Study selection and data extraction were independently performed by two investigators. Three reviewers solved any disagreement. We adopted the following steps in study selection: first, article selection based on the title and abstract, and second, full-text reading. Articles that did not meet the eligibility criteria were excluded from the review.

The following information extracted from the selected studies was written in an Excel 2019^®^ electronic form comprising the following fields: reference, title, source, journal, impact factor, study location, study design, follow-up period, and prevalence of obesity in Brazil, as well as in Brazilian regions and Brazilian capitals. We also extracted the sociodemographic characteristics of the population studied and the population with obesity, such as sex, race, urban or rural residence, and level of education.

### 2.6. Quality Assessment

Two investigators independently assessed the risk of bias in the selected studies according to the Joanna Briggs Institute (JBI) for assessing the risk of bias. The checklists evaluated were for analytical cross-sectional studies [15], cohort studies [15], and randomized controlled trials [16]. Disagreements were resolved in meetings by discussion among the three evaluators.

The overall certainty of the body of evidence was rated by using the Grading of Recommendations Assessment, Development, and Evaluation (GRADE) approach, taking into account the overall risk of bias, consistency of effect, imprecision, indirectness, and publication bias to assess the certainty of the body of evidence [17,18]. If there were serious concerns in any of these domains, we rated down the quality of evidence.

### 2.7. Statistical Analyses

This meta-analysis estimated obesity prevalence using the crude proportions method (PRAW) with random effects [19]. We chose this method because it corrected the overestimation of the weight of studies with estimates very close to 0% or 100% [19]. Subgroup analyses were performed by sex, Brazilian cities, regions, and housing location. Heterogeneity was assessed by the random-effects model, the chi-squared test was applied with a significance of *p* < 0.10, and its magnitude was determined by the I-squared (I^2^).

In all analyses, a *p*-value < 0.05 was considered statistically significant. Publication bias analysis was not performed as long as this measure is inappropriate for prevalence meta-analysis [20]. Analyses were performed in the RStudio software, version 4.1.0 (R: A Language and Environment for Statistical Computing, Vienna, Austria), by using the ‘Meta’ package, version 5.0-0.

## 3. Results

### 3.1. Studies Characteristics

Our search retrieved 5634 studies in the databases MEDLINE (via PubMed), Embase, Scopus, and LILACS. After excluding 1125 duplicates, 4509 titles and abstracts were screened. Full-text articles for the remaining 46 records were retrieved, of which 27 were excluded due to incorrect outcomes, like not describing the prevalence of obesity, an inappropriate population that is not representative of the Brazilian population, incorrect study design, or incomplete studies (Figure 1 and Appendix A).

The main characteristics of the included studies are summarized in Table 1.

Of the 40 included studies, 52.5% (*n* = 21) were primary studies, and 47.5% (*n* = 19) were secondary analyses (Figure 1). The PNSN study, Pesquisa Nacional de Saúde e Nutrição (National Survey on Health and Nutrition), was excluded from the meta-analysis, as cities in the interior of the northern region of Brazil were not included in the sample.

Brazil is a country with continental dimensions, divided into 26 states and a Federal District (Figure 2).

The states are grouped into five major geographic regions: North, Northeast, Southeast, South, and Midwest. The estimated Brazilian population for 2022 was 207.8 million inhabitants; the Southeastern region has the largest population contingent, followed by the Northeastern region, and the smallest population quantity is observed in the North and West-Central. The five Brazilian geographic regions show great socioeconomic inequality in terms of access to health. The best human development indices are observed in the South and the worst in the Northeastern and North regions, the latter being characterized by its low population density and territorial extension that shelters a large part of the Amazon rainforest. The Southeastern region, the most populous, is notable for its job market, whereas the West-Central, although it includes the capital of the country, has an economy primarily focused on agriculture and livestock [61].

The total population assessed in the studies was around 1.3 million people. Only 5.0% of the studies were carried out before the year 2000 (1975 and 1989), while 95.0% were developed between 2000 and 2023 (*n* = 38) and 40.0% (*n* = 18) in the last 5 years (Table 1). In the selected studies, obesity was defined as BMI > 30 kg/m^2^ in all studies. No other parameter was considered for classifying obesity.

### 3.2. Quality Assessment

Of the 40 studies included in the systematic review, 21 studies were evaluated and included in this meta-analysis. The Joanna Briggs Institute (JBI) instrument [62] for assessing the methodological quality of a systematic review of prevalence was used [63]. Studies were classified as low risk of bias if the total response was equal to 9, moderate risk of bias if between 6 and 8, and high risk of bias if ≤5. Regarding obesity outcome, we observed that 4 studies (20.0%) were identified as having a low risk of bias, 16 studies (80.0%) had a moderate risk of bias, and none had a high risk of bias (Table 2).

### 3.3. Meta-Analysis Results

In the evaluated studies, we did not find information on the planned secondary outcomes for the obese population. These secondary outcomes included the prevalence of obesity according to age group, race, skin color, level of education, income, and clinical variables such as chronic non-communicable diseases, self-assessed health conditions, and modifiable risk factors such as smoking, alcoholism, ingestion of ultra-processed foods and physical activity.

### 3.4. Obesity Prevalence in Brazilian’s Adulthood

The pooled prevalence of obesity in Brazilian adults in the national territory corresponds to 20.0% (95% CI: 14.0–25.0%), while in the capitals it was equal to 17.0% (95% CI: 16.0–19.0%) (Figure 3 and Figure 4). In the figures, the results are presented as a proportion; to facilitate understanding, we are describing them as a percentage value.

The prevalence of obesity was higher among women, at 20% (95% CI: 13.0–27.0%) in the national territory and 18% (95% CI: 16.0–19.0%) in the capitals, while in men it was 14% (95% CI: 7.0–20.0%) in the national territory and 17% (95% CI: 15.0–18.0%) in the capitals (Figure 5 and Figure 6).

In the regions of Brazil, the obese range is equivalent to a percentage between 11.0% and 17.0%, with the highest frequency in the south region (17.0%). Although, there was no statistical difference between the subgroups (*p* = 0.84) (Figure 7).

Obesity in the urban area was two percentage points higher than in the rural area, with a prevalence of obesity of 15.1% (95% CI: 3.1%; 27.2%) versus 12.7% (95% CI: 3.6%; 21.8%). However, there was no statistically significant difference between the subgroups (*p* = 0.75) (Figure 8).

The results of the meta-analysis of the prevalence of obesity in each Brazilian capital are summarized in Table 3 and presented in Appendix A.

In the capitals of the North region, the prevalence of obesity ranged from 13.8% (Palmas) to 19.6% (Manaus). When assessing the 95% confidence intervals, we observed significant differences between the prevalence of obesity in Palmas and the other capitals in northern Brazil. In the Northeastern capitals, the prevalence of obesity ranged from 13.7% (São Luís) to 18.4% (Fortaleza); we found a significant difference between the prevalence of obesity in the city of São Luís and the other capitals in the Northeast region (Table 3).

In the Southeast, the city of Belo Horizonte had the lowest (15.0%; 95% CI 13.4–16.7%) percentage of obesity, and Rio de Janeiro had the highest (18.6%; 95% CI 16.9–20.3%); it should be noted that this city also showed a higher prevalence when compared to Vitória (15.2%; 95% CI 13.9–16.5%). In the southern capitals, there was a significant difference between the prevalence of obesity in Porto Alegre (18.2%; 95% CI 16.6–19.7%) and Florianopolis (14.8%; 95% CI 13.5–16.1%) (Table 3). In the West-Central, we found a significant difference between the prevalence rates in Goiania (14.6%; 95% CI 12.8–16.3%) and the Federal District (14.6%; 95% CI 12.8–16.5%) when compared with those in Campo Grande (18.9%; 95% CI 17.3–20.7%) and Cuiabá (19.3%; 95% CI 17.5–21.2%) (Table 3).

## 4. Discussion

This review found a 20% overall prevalence of obesity among adults in Brazil, with a lower prevalence (17%) among residents of the capitals. The subgroup analysis by sex, geographic region of residence, and urban/rural residence showed no statistically significant differences. However, the analysis of state capitals by geographic region revealed differences in the prevalence of obesity between cities. The lowest percentages of obesity were observed in the following cities: Palmas had the lowest (13.8%) in the North region, São Luís in the Northeast (13.7%), Belo Horizonte in the Southeast (15.0%), Florianopolis in the South (15.0%), and Goiania (14.6%) and the Federal District (14.6%) in the Midwest. The summarized measures indicated high heterogeneity.

This is the first systematic review and meta-analysis summarizing data on obesity in Brazilian adults stratified by gender, the state capital, geographic region, and place of residence (rural and urban) extracted from extensive population-based surveys with probabilistic sampling. The main strength of this study is the low risk of bias observed. Systematically reviewing the literature and understanding the estimate of the prevalence of obesity among Brazilian adults is a critical way to make public health decisions aimed at reducing the prevalence of obesity in Brazil. Although there are reviews to summarize the prevalence of overweight (overweight and obesity) in Brazilian children, adolescents, and adults, the results have limited validity, primarily due to the lack of stratification and the focus on studies without national representation.

In all selected studies, the measure to assess nutritional status was the BMI proposed by the World Health Organization (WHO), providing greater validity to our findings. However, information on weight and height was self-reported; thus, the results are subject to information bias. Studies that evaluated the validity of self-reported weight and height with what was measured showed a tendency to reduce weight and increase height. Women showed a greater tendency to reduce weight and men to increase height. Results that may interfere with the BMI value are estimated through self-report, reducing its value and the prevalence of obese people in the population.

Another limitation concerns the survival bias that may appear in cross-sectional studies because when studying prevalent cases, factors associated with a greater or lesser probability of survival will interfere with the probability of being part of the study sample. Thus, as obese individuals are at greater risk of illness and death from NCDs and cancers compared to eutrophic individuals, they are less likely to be selected in random samples, not stratified by the nutritional status of the population. However, survival bias has a more significant impact on prevalence estimates over time, as certain individuals may be excluded from the data. Despite this, the estimated prevalence in this study is not affected by this bias.

The high heterogeneity observed in the summarized measures may be correlated with the fact that the studies are cross-sectional panel studies; that is, in each of the studies, we have a different sample of participants.

The prevalence of obesity observed in the Brazilian adult population in our meta-analysis in men (14.0%) and women (20.0%) was lower than that of the United States (31.6% vs. 33.9%) and higher than that of China (3.8% vs. 4.3%) and India (3.7% vs. 4.2%) [64].

Our estimates were higher than those presented by another study [65] carried out with four population-based studies (20.0 vs. 8.6%), a similar result when stratified by gender. However, our findings are close to those presented in the National Health Surveys of 2013 and 2019 [35].

Brazil is in an accelerated demographic, epidemiological, and nutritional transition process, showing an upward trend in the prevalence of overweight and obesity in adults, children, and adolescents [8,35,66]. According to data from Vigitel, from 2006 to 2021, there was a significant increase in the prevalence of obesity in all Brazilian capitals, in both sexes and adults. In the period from 2006 to 2021, the prevalence of obesity ranged from 11.8% to 22.4 (+89.8%), with an annual percentage variation of 0.66% (95% CI 0.57–0.74%). More significant percentage increases were observed in women (0.67%; 95% CI 0.59–0.76%) and in adults aged 35 to 44 years and with complete primary education (0.83%; 95% CI 0.76–0.91%) [8]. These results are similar to the global ones since there was a considerable increase in the prevalence of obesity in four decades, ranging from 3% to 11% among men and from 6% to 15% among women [64].

In this study, analyses by subgroup according to gender, geographic region, and place of residence did not show statistically significant differences. However, it is essential to highlight that women residing in Brazil and Brazilian capitals showed a higher prevalence of obesity than men, a trend like that observed in Brazilian population-based surveys and in other countries around the world [8,35,64,65,67,68]. In Brazil, there are differences in the sociodemographic profile between men and women with obesity. A higher prevalence of obesity is observed in men with higher per capita household income and a higher level of education; on the other hand, a higher prevalence of this health problem is found among women with low education and lower income [22,35].

Brazilian studies have shown a higher prevalence of obesity in men and women living in the South region and in men in the Midwest region. In addition, living in a rural area increased abdominal obesity (PR = 1.11; 95% CI 1.01–1.23) [29].

In this study, we found no differences in the prevalence of obesity between geographic regions, urban and rural areas. We believe that these results may be related to the fact that more than 80% of the studies were performed from 2006 to 2021.

In this historical period, Brazil had already become a country with a predominantly urban population, with significant changes in the pattern of food consumption and a reduction in the practice of physical activity associated with the expansion of fast-food chains, supermarket chains, agricultural production based on soy and corn monoculture, and reduction in the price of ultra-processed industrialized foods [8,69,70,71,72]. These socioeconomic and political changes may have affected all Brazilian regions, promoting a high prevalence of obesity even in Brazilian capitals that show a lower magnitude of this indicator.

We believe that this reality may worsen in the coming years due to the effects of the COVID-19 pandemic, as in Brazil, there has been an increase in the prevalence of unhealthy eating habits, a reduction in physical activity, and an increase in the consumption of alcoholic beverages and smoking [73,74,75].

Due to the increased prevalence and high morbidity and mortality, obesity is one of the greatest public health challenges, as it presents social depression and burdens health systems. These consequences of obesity and associated diseases are based on direct medical and non-medical costs, indirect costs such as lost productivity, and intangible costs such as decreased quality of life [76,77].

The direct costs attributable to obesity in Brazil in 2018 in the National Health Service were around BRL 378 million or USD 75 million. By adding obesity as a risk factor to NCDs such as systemic arterial hypertension and DM, we were able to obtain a more complete estimate of the economic impact of obesity on the SUS. The cost then increases to a total of BRL 1.39 billion [78], which is equivalent to approximately USD 278 million.

Challenges concern the interrelation between economic policy and the construction of a budget agreed upon among sectors. Integration into policy management and monitoring is necessary to promote intersectorality in the field of food and nutrition security and reduce obesity in the country [79].

## 5. Conclusions

The prevalence of obesity among adults of both sexes in Brazil and its states is high, with no statistically significant differences found in the subgroup analysis. This study found that the pooled prevalence of obesity in Brazilian adults corresponds to 20.0%. This alarming trend underscores the urgent need for comprehensive actions by the Brazilian government to address this public health issue.

Future research should focus on several key areas to better understand and combat obesity in Brazil. Long-term studies are essential to monitor changes in obesity prevalence and its determinants over time. This will help identify emerging trends and assess the effectiveness of implemented policies. Additionally, it is necessary to investigate the interaction between individual behaviors, environmental factors, and socioeconomic status in the development of obesity, as understanding these relationships can inform targeted interventions.

Evaluating the impact of existing policies aimed at reducing obesity, such as food labeling regulations, public health campaigns, and urban planning initiatives that promote physical activity, is also crucial. Comparative studies with other countries that have successfully reduced obesity rates can provide valuable insights.

Exploring the effectiveness of various nutritional interventions, including the promotion of healthy eating habits, access to fresh and affordable fruits and vegetables, and educational programs focused on nutrition literacy, is another important area of research. Similarly, it is essential to conduct detailed studies on the consumption patterns of ultra-processed foods and their association with obesity. Research should assess the impact of controlling the advertising of these foods and the potential benefits of increasing taxes on them as measures that have proven effective in other countries.

Obesity can also be influenced by genetic and epigenetic factors. Therefore, it is necessary to examine the role of these factors to identify individuals at higher risk and develop personalized intervention strategies. Furthermore, understanding the cultural and social influences on dietary habits and physical activity levels in different regions of Brazil can help design culturally sensitive and effective public health interventions.

Finally, investigating the role of the healthcare system in managing obesity, including the availability and accessibility of weight management programs, the integration of obesity prevention and treatment into primary care, and the training of healthcare professionals in obesity management, is essential for a comprehensive approach.

The high prevalence of obesity in Brazil signals the need for multifaceted and sustained efforts to curb this epidemic. Policies that promote access to healthy foods, control the advertising of ultra-processed foods and increase taxes on these foods should be prioritized, given their proven effectiveness in other countries. Future research must continue to build on these foundations, providing evidence-based solutions that are tailored to the unique challenges faced by the Brazilian population.

## Figures and Tables

**Figure 1 ijerph-21-01022-f001:**
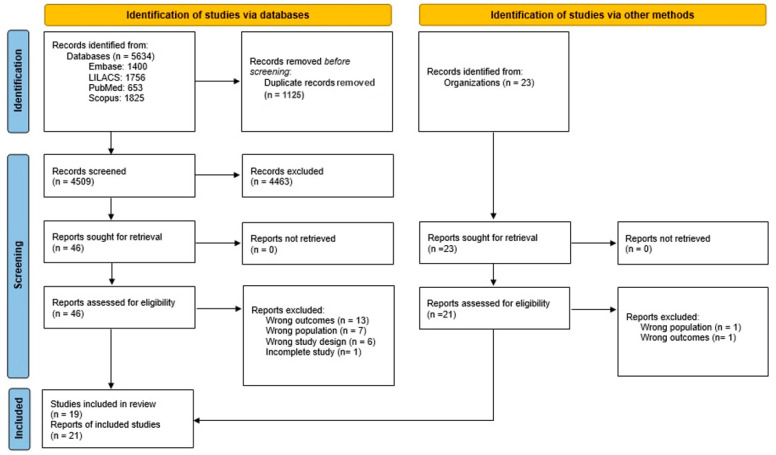
Study selection flowchart.

**Figure 2 ijerph-21-01022-f002:**
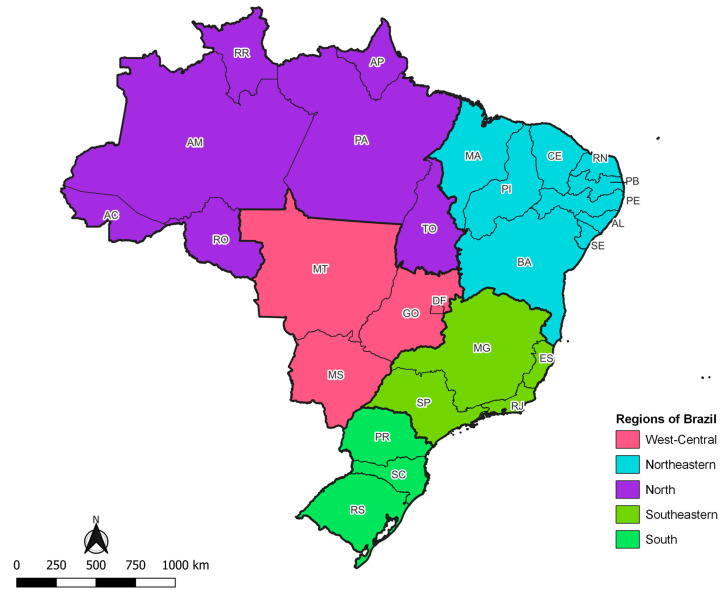
Map of Brazil, according to geographic region and states.

**Figure 3 ijerph-21-01022-f003:**
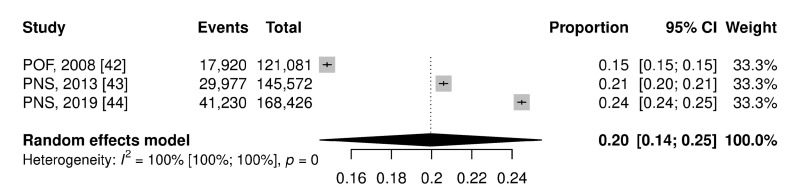
Forest plot of pooled prevalence ratio of obesity total on the national territory.

**Figure 4 ijerph-21-01022-f004:**
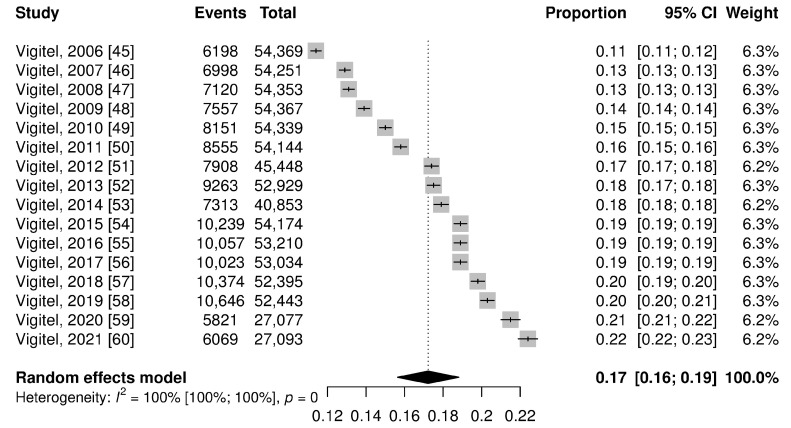
Forest plot of pooled prevalence ratio of obesity total on capitals.

**Figure 5 ijerph-21-01022-f005:**
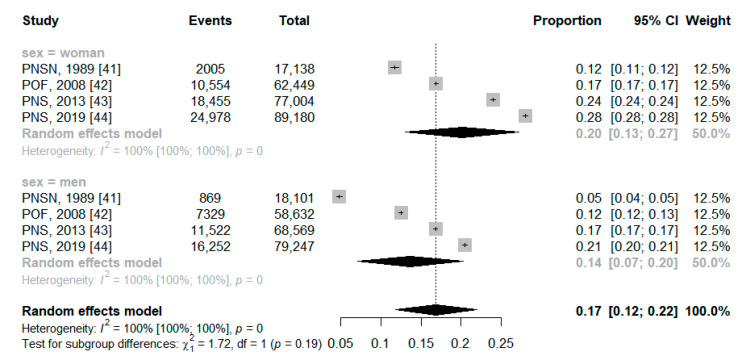
Forest plot of pooled prevalence ratio of obesity total on national territory by sex.

**Figure 6 ijerph-21-01022-f006:**
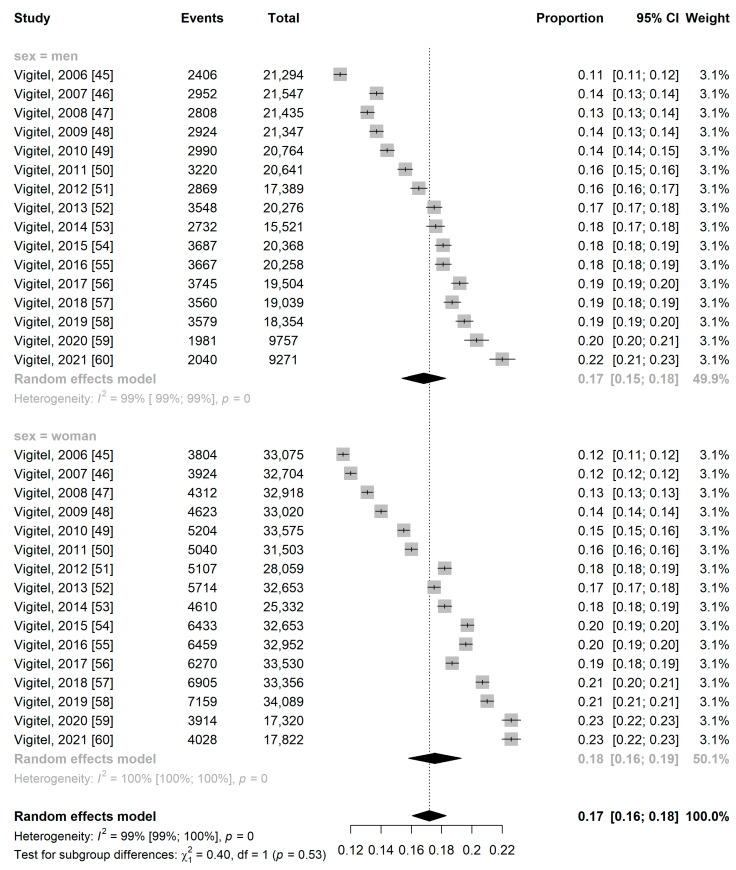
Forest plot of pooled prevalence ratio of obesity total on capitals by sex.

**Figure 7 ijerph-21-01022-f007:**
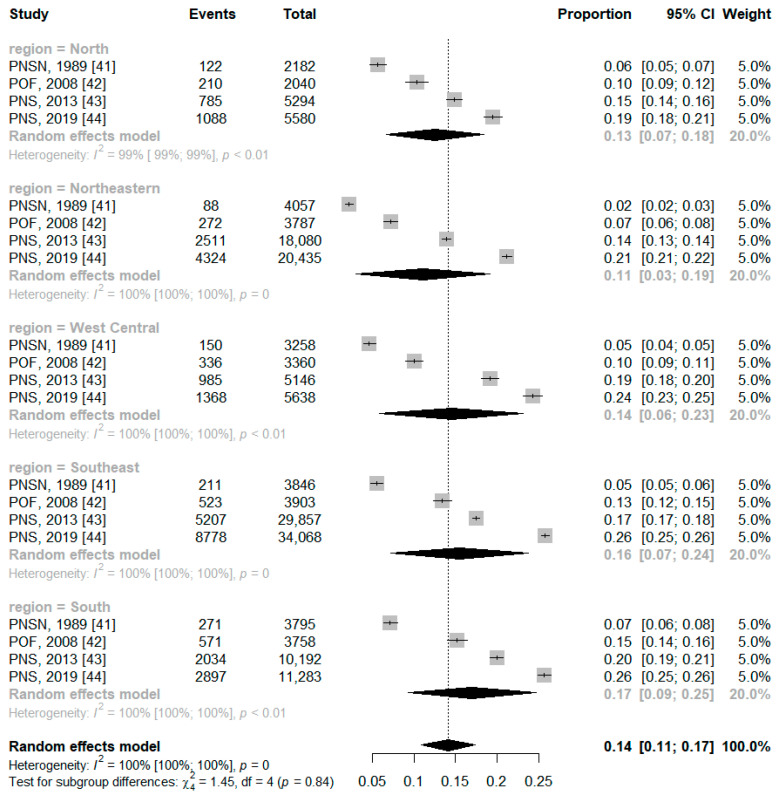
Forest plot of pooled prevalence ratio of obesity by Brazilian region.

**Figure 8 ijerph-21-01022-f008:**
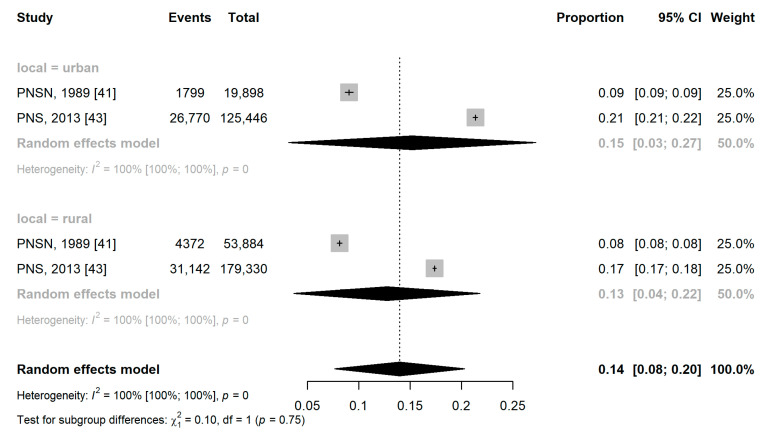
Forest plot of pooled prevalence ratio of obesity by housing location.

**Table 1 ijerph-21-01022-t001:** Main characteristics of included studies in meta-analysis.

Reference	Year	Journal	Study Design	Study Location	Sample Size	% Obesity	Women	% Women Obesity	Men	% Men Obesity
Monteiro et al., 2007 [21]	1974–1975	*American Journal of Public Health*	ENDEF ^a^	Brazil	124,274	-	62,709	7.4	57,179	2.7
	1979		PNSN ^a^	Brazil	32,651	-	15,827	12.4	15,435	5.1
Gigante et al., 2009 [22]	2006	*Revista de Saúde Pública*	Vigitel ^a^	Capitals	49,395	-	28,773	11.5	20,622	11.3
Gigante et al., 2011 [23]	2006	*Revista Brasileira de Epidemiologia*	Vigitel ^a^	Capitals	53,882	11.4	-	11.5	-	11.3
	2007				53,802	12.7	-	11.8	-	13.5
	2008				53,895	13.2	-	13.2	-	13.1
	2009				53,908	13.8	-	13.9	-	13.7
Moura et al., 2012 [24]	2006	*International Journal of Public Health*	Vigitel ^a^	Capitals	41,897	10.3	24,135	12.5	17,762	11.4
	2007				41,833	-	23,872	14.4	17,961	13.4
	2008				41,631	-	23,948	13.6	17,683	13.0
	2009				40,711	13.5	23,347	15.5	17,364	13.9
Malta et al., 2014 [25]	2012	*Epidemiologia e Serviços de Saúde*	Vigitel ^a^	Capitals	45,448	17.4	-	18.2	-	16.5
Moura et al., 2013 [26]	2010	*Salud(i)Ciencia*	Vigitel ^a^	Capitals	54,339	15.0	-	15.5	-	14.4
Louro, M.B.; 2016 [27]	2006	Master’s thesis	Vigitel ^a^	Capitals	54,369	11.8	-	-	-	-
	2007				54,251	13.2	-	-	-	-
	2008				54,353	13.7	-	-	-	-
	2009				54,367	14.3	-	-	-	-
	2010				54,339	15.1	-	-	-	-
	2011				54,140	16.0	-	-	-	-
	2012				45,448	17.3	-	-	-	-
	2013				52,929	17.5	-	-	-	-
	2014				40,853	17.9	-	-	-	-
Santos, I.K.S.; 2018 [28]	2006	Master’s thesis	Vigitel ^a^	Capitals	31,976	9.0	-	-	-	-
	2007				31,291	10.1	-	-	-	-
	2008				30,051	10.4	-	-	-	-
	2009				29,310	11.0	-	-	-	-
	2010				28,371	11.8	-	-	-	-
	2011				27,133	12.9	-	-	-	-
	2012				21,605	14.2	-	-	-	-
	2013				22,727	13.7	-	-	-	-
	2014				17,365	15.1	-	-	-	-
	2015				20,260	16.5	-	-	-	-
	2016				19,786	16.0	-	-	-	-
Martins-Silva et al., 2019 [29]	2013	*Revista Brasileira de Epidemiologia*	PNS ^a^	Brazil	59,226	20.7	33,306	24.3	25,920	16.8
Ortiz, R.J.F.; 2019 [30]	2006	Master’s thesis	Vigitel ^a^	Capitals	-	9.0	-	12.1	-	11.7
	2007				-	10.1	-	12.8	-	13.9
	2008				-	10.4	-	13.6	-	13.7
	2009				-	11.0	-	14.3	-	14.1
	2010				-	11.8	-	15.2	-	14.6
	2011				-	12.9	-	16.0	-	15.6
	2012				-	14.2	-	17.4	-	16.5
	2013				-	13.7	-	16.6	-	17.6
	2014				-	15.1	-	17.4	-	17.7
	2015				-	16.5	-	18.9	-	18.2
	2016				-	16.0	-	18.8	-	18.1
Canella et al., 2020 [31]	2008–2009	*Public Health Nutrition*	POF ^a^	Brazil	84,660	-	42,434	13.8	41,226	11.3
Felisbino-Mendes et al., 2020 [32]	2017	*Population Health Metrics*	Vigitel ^a^	Capitals	-	27.4	-	29.8	-	24.6
Passos et al., 2020 [33]	2008–2009	*Nutrition, Metabolism & Cardiovascular Diseases*	POF ^a^	Brazil	105,348	14.1	-	-	-	-
Streb et al., 2020 [34]	2015	*Ciência & Saúde Coletiva*	Vigitel ^a^	Capitals	35,448	-	21,069	19.1	14,379	18.3
Ferreira et al., 2021 [35]	2013	*Revista Brasileira de Epidemiologia*	PNS ^a^	Brazil	59,592	20.8	31,235	24.4	28,357	16.8
	2019				6672	25.9	3547	29.5	3125	21.8
da Silva et al., 2021 [36]	2006	*Ciência e Saúde Coletiva*	Vigitel ^a^	Capitals	-	11.9	-	12.2	-	11.4
	2007				-	13.3	-	13.0	-	13.6
	2008				-	13.7	-	13.9	-	13.4
	2009				-	14.3	-	14.7	-	13.9
	2010				-	15.1	-	15.6	-	14.4
	2011				-	16.0	-	16.5	-	15.5
	2012				-	17.4	-	18.2	-	16.5
	2013				-	17.5	-	17.5	-	17.5
	2014				-	17.9	-	18.2	-	17.6
	2015				-	18.9	-	19.7	-	18.1
	2016				-	18.9	-	19.6	-	18.1
	2017				-	18.9	-	18.7	-	19.2
	2018				-	19.8	-	20.7	-	18.7
	2019				-	20.3	-	21.0	-	19.5
da Silva et al., 2021 [37]	2006	*Epidemiologia e Serviços de Saúde*	Vigitel ^a^	Capitals	-	11.8	-	12.1	-	11.4
	2007				-	13.3	-	13.1	-	13.6
	2008				-	13.7	-	13.9	-	13.4
	2009				-	14.3	-	14.7	-	13.9
	2010				-	15.1	-	15.6	-	14.4
	2011				-	16.0	-	16.5	-	15.5
	2012				-	17.4	-	18.2	-	16.5
	2013				-	17.5	-	17.5	-	17.5
	2014				-	17.9	-	18.2	-	17.6
	2015				-	18.9	-	19.7	-	18.1
	2016				-	18.9	-	19.6	-	18.1
	2017				-	18.9	-	18.7	-	19.2
	2018				-	19.8	-	20.7	-	18.7
	2019				-	20.3	-	21.0	-	19.5
Bertuol et al., 2022 [38]	2017	*European Journal of Sport Science*	Vigitel ^a^	Capitals	31,489	18.5	18,795	-	12,694	-
Conde et al., 2022 [39]	1974–1975	*Cadernos de Saúde Pública*	ENDEF ^a^	Brazil	-	-	68,799	8.9	63,018	3.0
	1979		PNSN ^a^	Brazil	-	-	17,901	14.5	16,788	5.6
	2008–2009		POF ^a^	Brazil	-	-	65,689	18.6	62,025	13.1
	2013		PNS ^a^	Brazil	-	-	33,478	26.0	25,920	17.5
	2019		PNS ^a^	Brazil	-	-	3272	29.6	3298	21.9
ENDEF; 1974 [40]	1974–1975	IBGE	Primary study	Brazil	52,990	-	-	6.9	-	2.4
PNSN; 1989 [41]	1989	Ministério da Saúde	Primary study	Brazil	35,239	-	17,138	11.7	18,101	4.8
POF; 2008 [42]	2008–2009	IBGE	Primary study	Brazil	121,081	14.8	62,449	16.9	58,632	12.5
PNS; 2013 [43]	2013	IBGE	Primary study	Brazil	145,572	20.8	77,004	-	68,569	-
PNS; 2019 [44]	2019	IBGE	Primary study	Brazil	168,426	25.9	89,180	-	79,247	-
Vigitel; 2006 [45]	2006	Ministério da Saúde	Primary study	Capitals	54,369	11.4	33,075	11.5	21,294	11.3
Vigitel; 2007 [46]	2007	Ministério da Saúde	Primary study	Capitals	54,251	12.9	32,704	12.0	21,547	13.7
Vigitel; 2008 [47]	2008	Ministério da Saúde	Primary study	Capitals	54,353	13.1	32,918	13.1	21,435	13.1
Vigitel; 2009 [48]	2009	Ministério da Saúde	Primary study	Capitals	54,367	13.9	33,020	14.0	21,347	13.7
Vigitel; 2010 [49]	2010	Ministério da Saúde	Primary study	Capitals	54,339	15.0	33,575	15.5	20,764	14.4
Vigitel; 2011 [50]	2011	Ministério da Saúde	Primary study	Capitals	54,144	15.8	31,503	16.0	20,641	15.6
Vigitel; 2012 [51]	2012	Ministério da Saúde	Primary study	Capitals	45,448	17.4	28,059	18.2	17,389	16.5
Vigitel; 2013 [52]	2013	Ministério da Saúde	Primary study	Capitals	52,929	17.5	32,653	17.5	20,276	17.5
Vigitel; 2014 [53]	2014	Ministério da Saúde	Primary study	Capitals	40,853	17.9	25,332	18.2	15,521	17.6
Vigitel; 2015 [54]	2015	Ministério da Saúde	Primary study	Capitals	54,174	18.9	32,653	19.7	20,368	18.1
Vigitel; 2016 [55]	2016	Ministério da Saúde	Primary study	Capitals	53,210	18.9	32,952	19.6	20,258	18.1
Vigitel; 2017 [56]	2017	Ministério da Saúde	Primary study	Capitals	53,034	18.9	33,530	18.7	19,504	19.2
Vigitel; 2018 [57]	2018	Ministério da Saúde	Primary study	Capitals	52,395	19.8	33,356	20.7	19,039	18.7
Vigitel; 2019 [58]	2019	Ministério da Saúde	Primary study	Capitals	52,443	20.3	34,089	21.0	18,354	19.5
Vigitel; 2020 [59]	2020	Ministério da Saúde	Primary study	Capitals	27,077	21.5	17,320	22.6	9757	20.3
Vigitel; 2021 [60]	2021	Ministério da Saúde	Primary study	Capitals	27,093	22.4	17,822	22.6	9271	22.0

Table legend. PNSN, Pesquisa Nacional de Saúde e Nutrição (National Survey on Health and Nutrition); POF, Pesquisa de Orçamentos Familiares (Household Budget Survey); PNS, Pesquisa Nacional de Saúde (National Health Survey); Vigitel, Sistema de Vigilância de Fatores de Risco e Proteção para Doenças Crônicas por Inquérito Telefônico (Surveillance System for Risk and Protection Factors for Chronic Diseases by Telephone Survey); IBGE, Instituto Brasileiro de Geografia e Estatística (Brazilian Statistics Institute). ^a^ Secondary analysis of primary study.

**Table 2 ijerph-21-01022-t002:** Risk of bias assessment of included studies.

Study Item, Year	Year of Survey	Critical Appraisal ^a^									
		1	2	3	4	5	6	7	8	9	Final Score
ENDEF, 1974 [40]	1974–1975	1	1	1	0	0	1	1	1	0	7
PNSN, 1989 [41]	1989	1	1	1	1	1	1	1	1	1	9
POF, 2008 [42]	2008–2009	1	1	1	1	1	1	1	1	1	9
PNS, 2013 [43]	2013	1	1	1	1	1	1	1	1	1	9
PNS, 2019 [44]	2019	1	1	1	1	1	1	1	1	1	9
Vigitel, 2006 [45]	2006	1	1	1	1	1	1	0	1	1	8
Vigitel, 2007 [46]	2007	1	1	1	1	1	1	0	1	1	8
Vigitel, 2008 [47]	2008	1	1	1	1	1	1	0	1	1	8
Vigitel, 2009 [48]	2009	1	1	1	1	1	1	0	1	1	8
Vigitel, 2010 [49]	2010	1	1	1	1	1	1	0	1	1	8
Vigitel, 2011 [50]	2011	1	1	1	1	1	1	0	1	1	8
Vigitel, 2012 [51]	2012	1	1	1	1	1	1	0	1	1	8
Vigitel, 2013 [52]	2013	1	1	1	1	1	1	0	1	1	8
Vigitel, 2014 [53]	2014	1	1	1	1	1	1	0	1	1	8
Vigitel, 2015 [54]	2015	1	1	1	1	1	1	0	1	1	8
Vigitel, 2016 [55]	2016	1	1	1	1	1	1	0	1	1	8
Vigitel, 2017 [56]	2017	1	1	1	1	1	1	0	1	1	8
Vigitel, 2018 [57]	2018	1	1	1	1	1	1	0	1	1	8
Vigitel, 2019 [58]	2019	1	1	1	1	1	1	0	1	1	8
Vigitel, 2020 [59]	2020	1	1	1	1	1	1	0	1	1	8
Vigitel, 2021 [60]	2021	1	1	1	1	1	1	0	1	1	8

Note: ENDEF, Estudo Nacional de Despesa Familiar (National Survey on Household Expenses); PNSN, Pesquisa Nacional de Saúde e Nutrição (National Survey on Health and Nutrition); POF, Pesquisa de Orçamentos Familiares (Household Budget Survey); PNS, Pesquisa Nacional de Saúde (National Health Survey); Vigitel, Sistema de Vigilância de Fatores de Risco e Proteção para Doenças Crônicas por Inquérito Telefônico (Surveillance System for Risk and Protection Factors for Chronic Diseases by Telephone Survey). ^a^ Critical appraisal according to The Joanna Briggs Institute Critical Appraisal Checklist for Studies Reporting Prevalence Data: 1. Was the sample frame appropriate to address the target population? 2. Were study participants sampled in an appropriate way? 3. Was the sample size adequate? 4. Were the study subjects and the setting described in detail? 5. Was the data analysis conducted with sufficient coverage of the identified sample? 6. Were valid methods used for the identification of the condition? 7. Was the condition measured in a standard, reliable way for all participants? 8. Was there appropriate statistical analysis? 9. Was the response rate adequate, and if not, was the low response rate managed appropriately?

**Table 3 ijerph-21-01022-t003:** Results of the meta-analysis according to the capital of the Brazilian states.

Region ^a^	Capitals ^a^	Population	Sample	Proportion (95% CI)	I^2^ (%)	*p*-Value
North	Belém (PA)	29,270	4903	0.1714 [0.1530–0.1898]	93.9	<0.01
	Boa Vista (RR)	27,006	4509	0.1725 [0.1537–0.1913]	93.8	<0.01
	Macapá (AP)	26,802	4895	0.1874 [0.1712–0.2037]	91.3	<0.01
	Manaus (AM)	28,467	5468	0.1963 [0.1737–0.2188]	96.1	<0.01
	Palmas (TO)	28,835	3895	0.1382 [0.1219–0.1545]	94.5	<0.01
	Porto Velho (RO)	28,945	5349	0.1882 [0.1706–0.2058]	93.4	<0.01
	Rio Branco (AC)	28,156	5261	0.1904 [0.1713–0.2096]	95.4	<0.01
Northeastern	Aracaju (SE)	29,301	4991	0.1741 [0.1566; 0.1916]	93.2	<0.01
	Fortaleza (CE)	29,329	5310	0.1837 [0.1689–0.1986]	91.3	<0.01
	João Pessoa (PB)	29,248	5066	0.1758 [0.1586–0.1930]	93.6	<0.01
	Maceió (AL)	29,437	5163	0.1789 [0.1604–0.1974]	94.5	<0.01
	Natal (RN)	29,411	5079	0.1764 [0.1484–0.2045]	99.5	<0.01
	Recife (PE)	29,394	5069	0.1754 [0.1556–0.1952]	95.3	<0.01
	Salvador (BA)	29,306	4576	0.1584 [0.1417–0.1750]	94.1	<0.01
	São Luís (MA)	29,264	3978	0.1377 [0.1234–0.1520]	92.8	<0.01
Southeastern	Belo Horizonte (MG)	29,507	4368	0.1503 [0.1337; 0.1668]	93.9	<0.01
	Rio de Janeiro (RJ)	29,360	5399	0.1861 [0.1695–0.2027]	93.4	<0.01
	São Paulo (SP)	29,505	5008	0.1727 [0.1552–0.1902]	93.8	<0.01
	Vitória (ES)	29,347	4408	0.1519 [0.1390–0.1649]	90.6	<0.01
South	Curitiba (PR)	29,873	4934	0.1671 [0.1542–0.1800]	88.9	<0.01
	Florianópolis (SC)	29,123	4263	0.1482 [0.1354–0.1610]	89.9	<0.01
	Porto Alegre (RS)	29,383	5288	0.1817 [0.1664–0.1969]	92.3	<0.01
West-Central	Campo Grande (MS)	29,311	5520	0.1889 [0.1726–0.2072]	94.2	<0.01
	Cuiabá (MT)	29,344	5591	0.1932 [0.1745–0.2119]	94.5	<0.01
	Distrito Federal (DF)	29,251	4196	0.1468 [0.1279–0.1657]	95.3	<0.01
	Goiania (GO)	29,459	4240	0.1461 [0.1288–0.1634]	94.0	<0.01

Note: ^a^ meta-analysis of 16 studies using the crude proportions method (PRAW) with random effect.

## Data Availability

The data are available at: https://doi.org/10.5281/zenodo.13138708.

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
