# Peer review of "Obesity and Associated Factors in Brazilian Adults: Systematic Review and Meta-Analysis of Representative Studies"

_ijerph, 2024, doi:10.3390/ijerph21081022_

Round 1

Reviewer 1 Report

Comments and Suggestions for Authors

The article presents a systematic review and meta-analysis of representative studies conducted in Brazil to estimate the prevalence of obesity among Brazilian adults. The review was performed following PRISMA reporting guidelines. After applying the inclusion criteria, the authors interrogated five independent databases and retrieved 21 studies from the initial 5,643 identified.

The introduction is well-written and clearly presents state-of-the-art information regarding obesity, highlighting obesity-related pathology in Brazil.

The Materials and Methods section clearly explains the search strategy, primary and secondary outcomes, eligibility criteria, and the process for selecting studies and extracting data. This section also describes how the quality of the evaluation was ensured to reduce the risk of errors, as well as the methods used for statistical analysis.

The Results section is well-organized and schematically presents the study selection diagram and the main characteristics of the studies included in the meta-analysis. The map of Brazil and the information about the five major geographical regions help the reader understand the results, which are stratified by each region.

There are three possible drafting errors that require checking: Florianópolis 14.8% instead of 15.2% (line 304), Goiania CI 16.3% instead of 16.6% (line 306), and Cuiabá 19.3% instead of 18.9% (line 308).

The Discussion section reflects and explains the results within the current socio-economic context.  

The conclusions are relevant and draw attention to obesity as a global problem, providing appropriate solutions that have proven beneficial in other countries.  

The article addresses a problem of global interest and cites current sources, 60% of which were published in the last five years. The scientific rigor, accuracy of the data, and clarity of the presentation add significant value to the field of study, and therefore, I recommend it for publication after the three minor corrections mentioned above.

Author Response

Dear Reviewer,

Thank you very much for your thorough review and valuable feedback on our manuscript titled "Systematic Review and Meta-Analysis of Obesity Prevalence Among Brazilian Adults". We sincerely appreciate the time and effort you dedicated to providing insightful comments that have greatly improved the quality of our work.

We are pleased that you found the introduction to be well-written and informative, effectively presenting current insights into obesity and its implications in Brazil. Your positive feedback on the Materials and Methods section is also appreciated, acknowledging its clarity in describing our search strategy, criteria for study selection, and data extraction methods, all conducted in accordance with PRISMA guidelines.

Regarding the Results section, we are glad you found it well-organized and informative, particularly noting the study selection diagram and regional stratifications. We have duly noted the drafting errors you pointed out regarding the specific percentages in Florianópolis, Goiania, and Cuiabá, and we will correct these accordingly.

Furthermore, we are grateful for your positive assessment of the Discussion section, where we aimed to contextualize our findings within the socio-economic landscape. Your endorsement of the conclusions as relevant and impactful in addressing obesity as a global challenge, and your recognition of the scientific rigor and clarity of our presentation, are deeply appreciated.

We have made the necessary corrections as per your suggestions and believe these revisions have strengthened the manuscript. We are confident that our article contributes meaningfully to the literature on this important public health issue.

Once again, thank you for your constructive feedback and your recommendation for publication. We look forward to the final decision and hope our work will contribute to advancing the field.

Reviewer 2 Report

Comments and Suggestions for Authors

This manuscript is a review article, centered on the selection of population-based studies that are representative of Brazilian capitals or Brazil as a whole, in order to understand the estimate of the prevalence of obesity among Brazilian adults.  It is well presented, the research design is appropriate and the majority of relevant references are included. However, it is a study based on a single population cohort and the extracted conclusions could not be used as a base in order to extract information /data that could be adopted to other populations. Moreover, albeit the quality of presentation is adequate, the overall interest and merit to the readers is restricted. This is mainly due to the fact that this research is unable to add something new in the scientifc community.

Comments on the Quality of English Language

The quality of English language is adequate, however minor corrections by a native English speaker seem to be necessary.

Author Response

Dear Reviewer,

Thank you for your thoughtful review and constructive feedback on our manuscript titled "Review Article on the Prevalence of Obesity Among Brazilian Adults." We appreciate the time and effort you invested in providing us with valuable insights.

We acknowledge your positive assessment of the presentation and research design of our review, as well as the inclusion of relevant references. Your observations regarding the study being based on a single population cohort and its limitations in generalizability to other populations are duly noted. 

We also appreciate your feedback on the adequacy of the English language quality. We Will make the necessary corrections to enhance the clarity and coherence of the manuscript.

Regarding the concern about the manuscript's contribution to the scientific community, we understand the importance of adding novel insights. In our revision, we will strive to better highlight the implications and relevance of our findings within the broader context of obesity research.

Once again, we thank you for your valuable comments, which will undoubtedly help us improve the quality and impact of our manuscript. We look forward to submitting the revised version and hope it meets your expectations.

Reviewer 3 Report

Comments and Suggestions for Authors

This study reviews a significant body of research on obesity in the Brazilian population. A significant number of sources include the construction of a systematic review in accordance with the basic principles of Preferred Reporting Items for Systematic Reviews and Meta-Analyses (PRISMA), and thorough conclusions based on the results of the analysis are impressive in this study. The topic discussed by the authors is undoubtedly an actual problem of modern life not only for Brazil.
The methods used in the study fully correspond to the purpose of the study and provide its solution.
The authors pay considerable attention to the researches of recent years, what allow determine the modern tendency in scientific research of obesity.
The presentation of the material allows you to fully analyze the progress of the research.
Regarding comments on this scientific work:
The abstract should present the main criteria for inclusion and exclusion of scientific publications in the system review.
From the point of view of a reader, it was interesting to provide links to the publications that were subject of the final analysis for the possibility of detailed familiarization if necessary.
The conclusions should be revised and expanded taking into account both the obtained results and the directions of future research on this issue.

Author Response

Dear Reviewer,

Thank you very much for your thoughtful review and valuable feedback on our study titled "Review of Obesity Research in the Brazilian Population." We appreciate the time and effort you dedicated to providing us with insightful comments.

We are pleased that you found our systematic review adhering to the principles of Preferred Reporting Items for Systematic Reviews and Meta-Analyses (PRISMA) and noted the thorough conclusions drawn from the analysis. The topic we addressed is indeed a pressing issue in modern life, not only in Brazil but globally.

We are glad that the methods employed in our study align with its purpose and effectively contribute to addressing the research questions. The emphasis on recent research trends in obesity allows us to identify current developments in the field.

Your suggestions regarding the abstract, including clearer presentation of inclusion and exclusion criteria, and providing links to publications for further reading, are well-received. We will incorporate these recommendations into the revised manuscript. Additionally, we appreciate your feedback on expanding the conclusions to encompass both our findings and future research directions on this important issue.

Once again, thank you for your constructive comments, which will undoubtedly help us enhance the clarity and impact of our study. We look forward to submitting the revised manuscript and hope it meets your expectations.

Round 2

Reviewer 2 Report

Comments and Suggestions for Authors

The authors attempted, by submitting the revised version of their manuscript, to eliminate limitations of their study, centered on the presentation and research design of their review, as well as the inclusion of relevant references.

However, the main drawback of this study is that it lacks any significant contribution to the scientific community, as one of the most important points of any such article is the addition of any novel insights.

Comments on the Quality of English Language

I would like to comment that only minor editing of English language is required.

Author Response

Thank you for reviewing our manuscript and providing detailed feedback. We appreciate your insights and would like to address your concerns regarding the significance and novelty of our study.

Our study represents the first systematic review and meta-analysis that synthesizes data on the prevalence of obesity among Brazilian adults, evaluated through extensive population-based surveys with probabilistic sampling. Although previous reviews have focused broadly on overweight and obesity across different age groups in Brazil, our study adds value by providing a stratified analysis of obesity specifically across various sociodemographic parameters such as gender, geographic region, and urban versus rural residence. 

We acknowledge that our findings regarding prevalence rates (e.g., 20% global, 17% in state capitals) align closely with existing public health data. However, our stratified approach offers unique insights into regional disparities and specific sociodemographic trends that have not been comprehensively analyzed before. For instance, we observed variations in obesity prevalence rates among different state capitals, with cities like Palmas and São Luís showing significantly lower rates. This granular level of detail is critical for tailoring public health interventions to local needs, offering a more targeted approach to combating obesity.

Here are a few points to emphasize our study’s contribution:

1. Stratified Data Analysis:
   - By breaking down obesity prevalence according to sex, geographic region, and place of residence, our study illuminates patterns that could inform regional public health strategies. Identifying that there is no statistically significant difference in prevalence between urban and rural areas, or among geographic regions, suggests that obesity is a pervasive issue requiring nationwide policy interventions.

2. High-Quality Data with Probabilistic Sampling:
   - All included studies utilized BMI as per WHO standards, enhancing the validity of our results. We also ensured that there was no high risk of bias in our selected studies, which strengthens the reliability of our findings.

3. Unique Timeframe:
   - Our analysis covers data from 2006 to 2021, a period during which Brazil underwent significant sociodemographic and nutritional transitions. Examining data from this timeframe provides a snapshot of obesity trends during a critical period of change, offering insights that can guide future public health initiatives.
4. Public Health Implications:
   - Understanding the estimated prevalence of obesity and its sociodemographic determinants is crucial for developing effective public health policies. Our study suggests that although obesity prevalence varies little by region or urban/rural status, comprehensive, multifaceted public health strategies are needed to address the issue nationwide.
5. Economic Impact:
   - By highlighting the direct and indirect costs associated with obesity, our study underscores the importance of investing in preventive measures. Given the rising economic burden, our research supports the need for policy interventions that promote healthier lifestyles and reduce obesity prevalence.
Future Research Directions:
To further build on our findings, we propose the following areas for future research:
- Longitudinal studies to monitor trends and assess the impact of implemented policies over time.
- Research exploring the interaction between individual behaviors, environmental factors, and socioeconomic status.
- Evaluation of the effectiveness of existing public health interventions aimed at reducing obesity.
- Detailed study of consumption patterns of ultra-processed foods and their regulation.
- Investigation into genetic and epigenetic factors affecting obesity prevalence.
- Qualitative studies on cultural and social influences on diet and physical activity.
- Examination of the healthcare system’s role in managing obesity, with a focus on accessibility and integration into primary care.
In conclusion, we assert that our study provides critical, detailed information on obesity patterns in Brazil, filling a gap in the literature and offering actionable data for policymakers. Our contributions, while perhaps not groundbreaking in terms of novel findings, are significant in terms of their potential to inform effective public health strategies and address a pressing healthcare issue in Brazil.
Thank you once again for your constructive feedback. We hope that our clarifications address your concerns, and we welcome any further suggestions to enhance our manuscript.

We have also sent the letter with proof of the foreign language we pay for our work.
